# A Carbazole-Functionalized Porous Aromatic Framework for Enhancing Volatile Iodine Capture via *Lewis* Electron Pairing

**DOI:** 10.3390/molecules26175263

**Published:** 2021-08-30

**Authors:** Zhuojun Yan, Bo Cui, Ting Zhao, Yifu Luo, Hongcui Zhang, Jialin Xie, Na Li, Naishun Bu, Ye Yuan, Lixin Xia

**Affiliations:** 1College of Chemistry, Liaoning University, Shenyang 110036, China; zjyan@lnu.edu.cn (Z.Y.); cuibo2019@163.com (B.C.); zhanghongcui2021@126.com (H.Z.); xjl810888632@163.com (J.X.); linaa627@163.com (N.L.); 2School of Environmental Science, Liaoning University, Shenyang 110036, China; zhaot17865818271@163.com (T.Z.); royjacklyf@126.com (Y.L.); 3Key Laboratory of Polyoxometalate and Reticular Material Chemistry of Ministry of Education, Faculty of Chemistry, Northeast Normal University, Changchun 130024, China; 4Yingkou Institute of Technology, Yingkou 115014, China

**Keywords:** iodine capture, porous aromatic framework, *Lewis* electron, pairing effect, Sonogashira-Hagihara cross-coupling

## Abstract

Nitrogen-rich porous networks with additional polarity and basicity may serve as effective adsorbents for the *Lewis* electron pairing of iodine molecules. Herein a carbazole-functionalized porous aromatic framework (PAF) was synthesized through a Sonogashira–Hagihara cross-coupling polymerization of 1,3,5-triethynylbenzene and 2,7-dibromocarbazole building monomers. The resulting solid with a high nitrogen content incorporated the *Lewis* electron pairing effect into a π-conjugated nano-cavity, leading to an ultrahigh binding capability for iodine molecules. The iodine uptake per specific surface area was ~8 mg m^−2^ which achieved the highest level among all reported I_2_ adsorbents, surpassing that of the pure biphenyl-based PAF sample by ca. 30 times. Our study illustrated a new possibility for introducing electron-rich building units into the design and synthesis of porous adsorbents for effective capture and removal of volatile iodine from nuclear waste and leakage.

## 1. Introduction

To overcome the energy shortages and environmental concerns originated from fossil fuels, nuclear power, the only mature technology, is considered a possible approach for providing electricity on a large scale with little greenhouse gases emission [1]. However, the treatment of nuclear waste and the emergency response for nuclear leakage, cause consternation in the increasing development of the nuclear industry [2]. The ^129^I and ^131^I atoms originated from the uranium fission are the two main ingredients of nuclear waste, especially ^129^I, which has an ultra-long radioactive half-life (t_1/2_ = 15.7 × 10^6^ years) [3,4]. Because the enrichment and toxic effects in organisms, effective methods for the capture and removal of radiological iodine aroused strong concerns. To date, several strategies have been proposed, including dry dedusting [5,6], chemical precipitation [7], and physical adsorption [8,9,10]. Among them, the physical adsorption method has specific advantages of high adsorption efficiency, low cost, simple operation, and high recyclability [11,12].

Porous aromatic frameworks (PAFs) composed of covalently bonded light atoms (H, B, C, N, and O), have superb thermal and chemical stability, high surface area, and tunable pore size, which make them ideal candidates for iodine capture from the nuclear waste stream containing volatile iodine radionuclides [13,14,15,16]. In the past few decades, PAF solids with tunable pore properties including surface area, volume, and size distribution were demonstrated to play important roles for the physical adsorption for guest molecules [17,18,19,20]. However, pure carbon-based PAFs with a micropore cavity do not show an excellent capacity and fast kinetics for I_2_ matter adsorption. For instance, PAF-1 with an exceptionally high surface area (5600 m^2^ g^−1^) and micropore volume (0.89 cm^3^ g^−1^) exhibits a low iodine vapor capture capability with 186 wt% at 298 K per 40 Pa [21]. It is obvious that the adsorption capacity of the adsorbent for iodine is not only related to the surface area and pore size, but the effective adsorption sites on the accessible surface may possess a more important role to interact with volatile iodine gases. A detailed investigation should be conducted to reveal the relationship between the chemical features of PAFs and iodine molecules, which provides significant advantages and opportunities of PAFs for the development of next-generation porous adsorptions.

Based on the polarization effect, active sites transform the speciation of iodine molecules into multiple oxidation states (−1, 0, +1, +3, +5, and +7), primarily as molecular iodine (I_2_), iodide (I^−^), iodate, or organic iodine (org-I) [22,23,24,25]. A nitrogenous fragment possesses lone pair electrons, thereby revealing highly negative charge to enhance the binding affinity for the polarizable electron cloud of I_2_ molecules [26]. Herein, 2,7-dibromocarbazole was adopted as the functional building monomer to prepare a carbazole-containing PAF network through a one-step Sonogashira-Hagihara coupling reaction. Consequently, the resulting PAF sample with the electron-rich system exhibits an outstanding performance for the capture of a volatile iodine with an uptake of 2.10 g g^−1^. The results of this study provide useful guidance for the development of new porous adsorbents for the removal of radioactive iodine.

## 2. Results and Discussion

LNU-13 was synthesized through the Sonogashira-Hagihara coupling of 2,7-dibromocarbazole and 1,3,5-triethynylbenzene (Figure 1a). As determined by the Fourier transform infrared spectroscopy (FTIR, Figure 2a), the C-Br stretching vibration of 2,7-dibromocarbazole at 495 cm^−1^ and the C–H stretching vibration of the terminal alkyne (1,3,5-triethynylbenzene) at 3270 cm^−1^ disappeared from the IR spectrum of LNU-13, verifying the completeness of the Sonogashira-Hagihara coupling reaction. The structural integrity of LNU-13 was further confirmed by ^13^C NMR (Figure 2b). The main peaks observed in the range of 120–150 ppm were attributed to the substituted carbon of the aromatic ring connected to the benzene ring; and the resonance around 90 ppm was assigned to the carbons originated from the –C≡C– group.

Powder X-ray diffraction (XRD) pattern of LNU-13 shows a characteristic broad peak, indicating they are amorphous in nature (Figure 2c). It seems that the formation of the stacked layer structure by the ordered connection among the building blocks is otherwise difficult [13,27]. Scanning electron microscopy (SEM) analysis demonstrated the stacked spherical structures of LNU-13, as shown in Figure 2d. Transmission electron microscopy (TEM) clearly confirmed the amorphous structure of LNU-13 (Figure 2e). As illustrated by thermogravimetric analysis (TGA, Figure 2f), the LNU-13 material begins to degrade at 350 °C and the weight loss is about 20% at 750 °C under a purified nitrogen atmosphere, indicating that LNU-13 possesses good thermal stability. All the results demonstrate that LNU-13 retains its intact skeleton under a variety of harsh conditions. 

The porosity of the resulting PAF material was probed using N_2_ adsorption-desorption isotherms at 77 K up to 1 bar. The adsorption curve combined the features of type-I and type-IV adsorption isotherms, indicating the co-existence of a micro- and meso-pore system (Figure 2g). The BET surface area of LNU-13 was determined to be 255 m^2^ g^−1^. LNU-13 possessed wide pore size distribution in the range of 1–6 nm calculated using a nonlocalized DFT (NL-DFT) (Figure 2h). This hierarchical porous structure made the PAF solid an excellent scaffold for the access of the I_2_ guest into the internal space of LNU particle [28,29].

The iodine uptake measurement of LUN-13 was conducted by placing the PAF powder into a sealed vessel filled with iodine vapor at 348 K under normal atmosphere. As shown in Figure 3a, the iodine adsorption capacity increased significantly with the prolonging of the contact time. In the first 5 h, the adsorption capacity of LNU-13 was very fast with a value of 1.75 g g^−1^. No further change in iodine loading was observed after 48 h exposure, indicating that LNU-13 was basically saturated (2.10 g g^−1^). A significant color change in the powder from brown to black was observed (Figure 3a inset). Calculated by the BET surface area (255 m^2^ g^−1^), the iodine uptake per specific surface area was ~ 8 mg m^−2^ which achieved the highest level among silver-containing zeolite [30], metal-organic frameworks (MOFs), and conjugated microporous polymers (CMPs), etc., reported by the same adsorption method, surpassing that of PAF-1 by ca. 30 times (Figure 4). Moreover, it also has a certain competitiveness compared with other forms of adsorbent, such as carbon foam, fiber adsorbent, carbon cloth, aerogel, etc., including BN foam (2.12 g g^−1^) [31], PE/PP-g-PNVP fibers (1.2378 g g^−1^) [32], C60-CC-PNP (2.4 g g^−1^) [33], CC-PNP (1.02 g g^−1^) [33], ENTDAT dried gel (1.8 g g^−1^) [34], G-TP5 (0.67 g g^−1^) [35] and G-TP6 (0.58 g g^−1^) [35].

The adsorption mechanism of iodine vapor in LNU-13 was studied through PXRD, Raman, and FT-IR spectroscopy. Curve-fitting for the I_2_ adsorption isotherm was based on pseudo-second-order kinetics (Figure 3b), a high correlation coefficient (*R*^2^ = 0.99993) suggested the chemical adsorption process of LNU-13. As shown in Figure 5a, there were no characteristic peaks of I_2_ crystal diffraction peaks observed in the iodine-loaded LNU-13 (LNU-13@I_2_). This phenomenon proved the monodispersed iodine species in the form of molecular or ionic states in the PAF architecture [27]. Raman spectroscopy of LNU-13@I_2_ presented a series of bands centered at 110 and 170 cm^−1^ (Figure 5b). The characteristic bands in the region of 100–120 cm^−1^ were assigned to the symmetric stretching of the I^3−^ species, while the band located at 170 cm^−1^ was ascribed to the higher polyiodide anions, i.e., I^5−^ [36,37]. Comparing the FTIR spectra of pristine LNU-13 and LNU-13@I_2_ (Figure 5c,d), the aromatic rings were centered at 1555 cm^−1^ in LNU-13 vs. 1612 cm^−1^ in I_2_@LNU-13. A similar shift was also observed for the band assigned to *v*_C–N_ (str) bond vibration (1234 cm^−1^ for LNU-13 and 1262 cm^−1^ for I_2_@LNU-13). In addition, the peak at 731 cm^−1^ belonged to the characteristic signal for iodine molecules. All these results indicate that the lone pair electron of the carbazole nitrogen polarizes the iodine molecule into an ionic state, and then achieves the excellent adsorption property for an iodine guest [38,39].

In order to evaluate the ability of LNU-13 for the capture of elemental iodine from the solution, LNU-13 powder was immersed into a closed vial containing a pre-prepared iodine elemental n-hexane solution (300 mg L^−1^). As depicted in Figure 3c, the color of the initial solution originated from iodine elemental substance changed from purple to colorless over time; after exposure for 24 h, the n-hexane solution containing both LNU-13 and iodine molecules became transparent and colorless, which proved that LNU-13 powder captured iodine from a *n*-hexane solution. 

The recyclability for I_2_ capture is also a key parameter in practical usage. The iodine-loaded LNU-13 powder can be activated by both thermal desorption and solvent elution. The iodine adsorbed in the PAF cavity is easily released in polar organic solvents including methanol and ethanol. After immersion in an ethanol solution for 72 h, the color of the mixture gradually changes from colorless to dark brown, correspondingly, the color of the solid varies from black to brown (Figure 3d). These results manifest that guest iodine is gradually released from the PAF structure into the organic solvent. As shown in Figure 3e, the release efficiency of LNU-13@I_2_ is as high as 97% after the solid is heated in air at 398 K for 320 min. In addition, the LNU-13 sample withstands multiple adsorption-desorption cycles, and the adsorption capacity reaches 69% of the initial capacity after five cycles of iodine adsorption (Figure 3f).

## 3. Materials and Methods

### 3.1. Materials

2,7-Dibromocarbazole was purchased from Energy Chemical, Shanghai, China and 1,3,5-triethynylbenzene was received from TCI, Tokyo, Japan. Copper iodide and tetrakis (triphenylphosphine) palladium were obtained from Sigma-Aldrich, St. Louis, MO, USA. Other chemicals and solvents were purchased from commercial suppliers and used without further purification. All reactions were performed under a purified nitrogen atmosphere.

### 3.2. Synthesis of LNU-13

The 2,7-Dibromocarbazole (649 mg, 1.9976 mmol), 1,3,5-triethynylbenzene (200 mg, 1.3317 mmol), tetrakis (triphenylphosphine) palladium (30 mg), and copper (I) iodide (10 mg) were added into a round-bottom flask. The mixture was degassed through a N_2_ bubbling process for 30 min; after that, 20 mL of anhydrous *N*,*N*-dimethylformamide (DMF) and 8 mL of anhydrous triethylamine (TEA) were added into the system. Then, the reaction mixture was heated to 80 °C for 72 h under N_2_ gas atmosphere. Cooling to room temperature, the precipitate was washed with each DMF, tetrahydrofuran (THF), and acetone solvents for several times to obtain a crude product. Further purification of the product was carried out via Soxhlet extraction with THF, dichloromethane, and methanol in turns for 72 h. The product was dried in a vacuum for 10 h at 90 °C to obtain LNU-13.

### 3.3. Iodine Adsorption and Release

#### 3.3.1. Iodine Adsorption from Volatile Iodine

The iodine adsorption capacity was analyzed according to the gravimetric measurements. The LNU-13 powder (30.0 mg) was loaded into a small weighing bottle, which was then placed in a closed system at 348 K (75 °C) and ambient pressure, along with excess non-radioactive solid iodine. After certain time intervals, the bottle was taken out, cooled down to room temperature and weighted, and then loaded back into the vapor of iodine to continue iodine adsorption [40,41]. The weight percentage of captured iodine was calculated using the following formula:(1)$$\mathrm{Adsorption}\;\mathrm{capacity}=\frac{m_{2}-m_{1}}{m_{1}}\times 100\%$$
where *m*_2_ and *m*_1_ are the masses of PAF powder after and before iodine intake, respectively.

#### 3.3.2. Iodine Adsorption from Solution

To evaluate the adsorption of dissolved iodine in cyclohexane, LNU-13 samples were immersed in n-hexane solution (300 mg L^−1^) containing iodine for 24 h, the adsorption process of iodine was photographed at selected time intervals.

#### 3.3.3. Iodine Desorption in Solution

Ethanol was used as the extraction solvent to evaluate the reversibility of PAF materials iodine adsorption. Pouring five milliliters of ethanol to five milligrams of iodine-loaded polymer, the release process of iodine was photographed at selected time intervals.

## 4. Conclusions

In summary, a carbazole-based porous aromatic framework was successfully synthesized through a one-step Sonogashira-Hagihara cross-coupling polymerization. Based on the *Lewis* electron pairing effect, the resulting solid achieved the highest value of iodine uptake per specific surface area. The iodine uptake per specific surface area far surpassed that of silver-containing zeolite, MOFs, and CMPs, etc. Our study firmly demonstrated the important role of electron-rich units in the open architecture for capture and the removal of iodine substance, which opened a gate for the design and synthesis of porous adsorbents for remediation of radioactive iodine to address environmental issues.

## Figures and Tables

**Figure 1 molecules-26-05263-f001:**
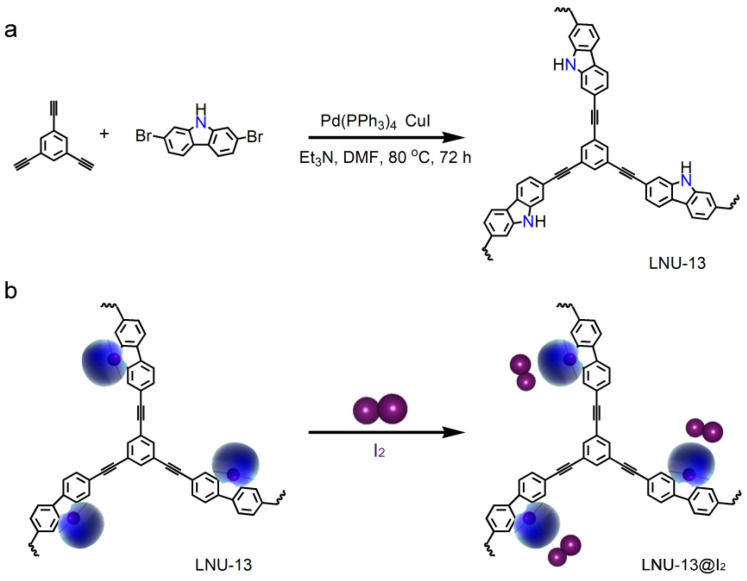
(**a**) Synthesis of LNU-13 polymer; (**b**) schematic diagram of PAF solid for I_2_ sorption.

**Figure 2 molecules-26-05263-f002:**
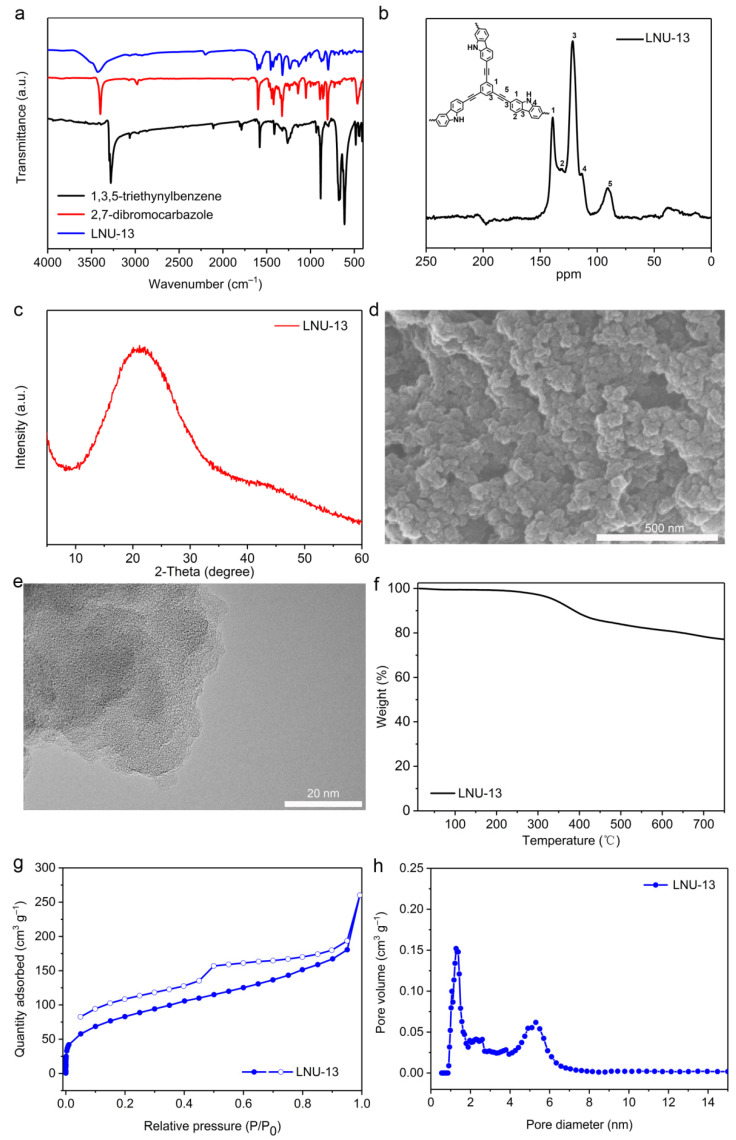
(**a**) FTIR spectra of 2,7-dibromocarbazole, 1,3,5-triethynylbenzene, and LNU-13; (**b**) solid-state ^13^C NMR spectrum of LUN-13; (**c**) powder X-ray diffraction pattern of LNU-13. (**d**) SEM image of LNU-13; (**e**)TEM image of LNU-13; (**f**) TGA plot of LNU-13 at N_2_ condition with a ramp rate of 5 °C min^−1^; (**g**) N_2_ adsorption-desorption isotherm of LNU-13; (**h**) pore size distribution of LNU-13.

**Figure 3 molecules-26-05263-f003:**
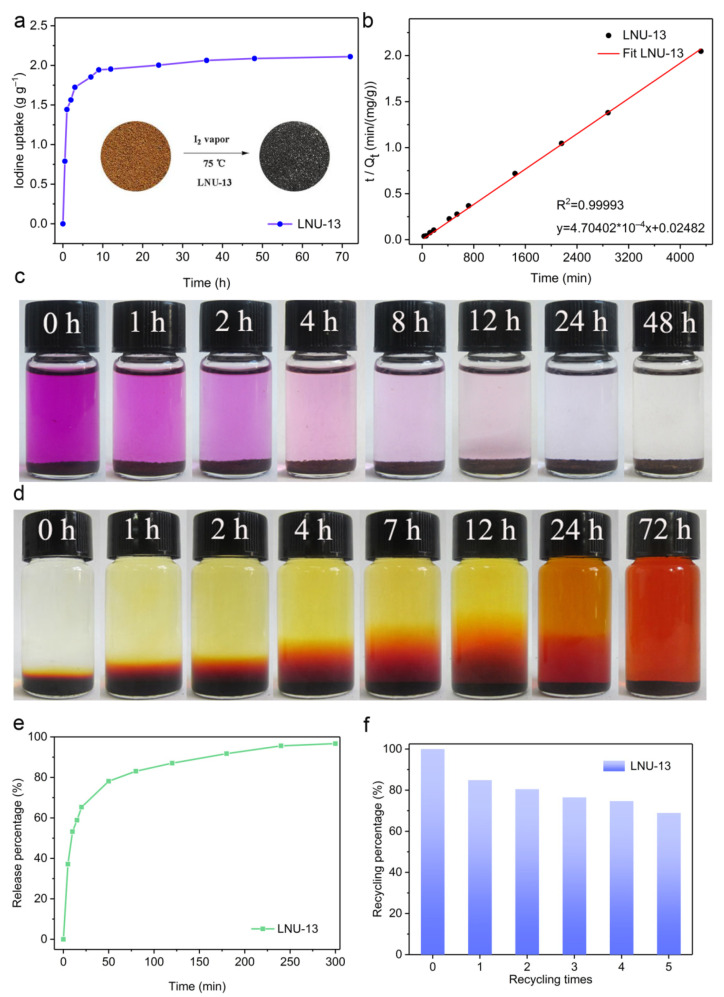
(**a**) I_2_ adsorption curve of LNU-13 at 348 K. Inset: the photographs reveal the color change in LNU-13 before and after iodine adsorption; (**b**) curve-fitting for the I_2_ adsorption process; (**c**) photographs showing the iodine-adsorbed process in n-hexane; (**d**) photographs showing the iodine-released process of LNU-13@I_2_ in ethanol; (**e**) I_2_ release curve of LNU-13@I_2_ at 398 K; (**f**) recycling experiment of LNU-13.

**Figure 4 molecules-26-05263-f004:**
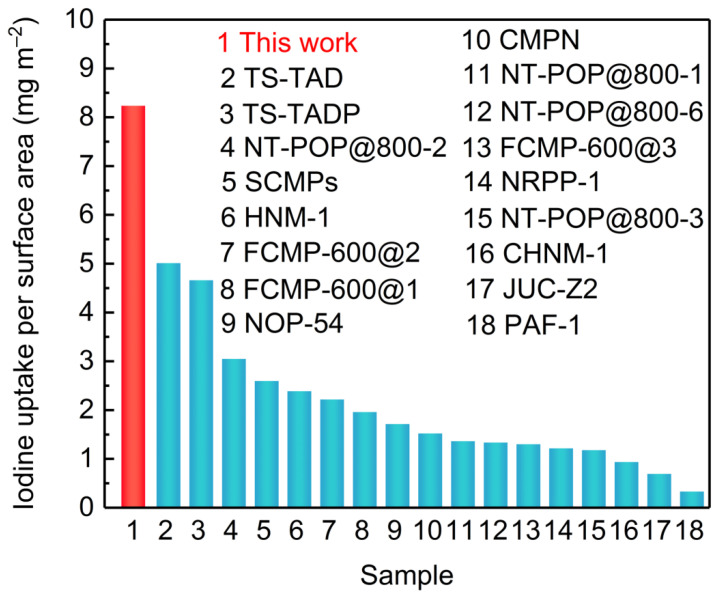
Iodine uptake capacities of different adsorbents.

**Figure 5 molecules-26-05263-f005:**
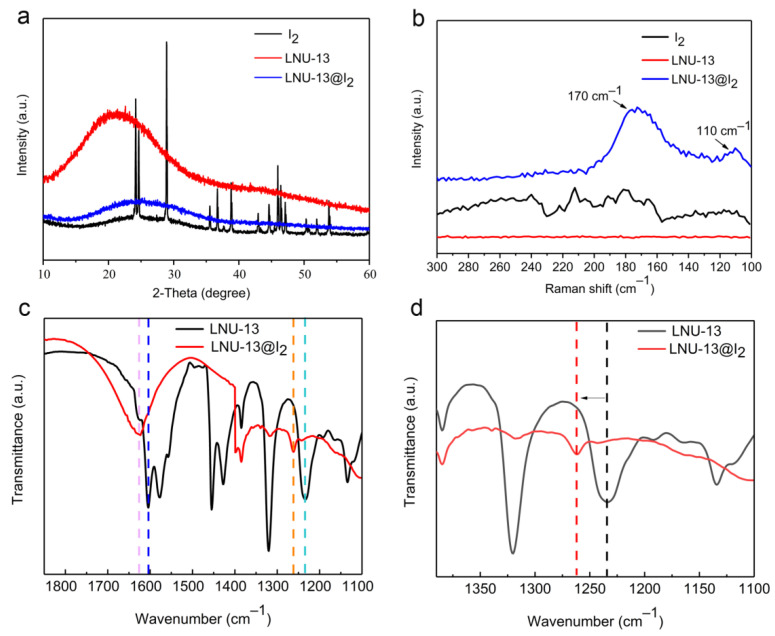
(**a**) PXRD spectra of I_2_, LNU-13, and LNU-13@I_2_; (**b**) Raman spectra of I_2_, LNU-13, and LNU-13@I_2_; (**c**,**d**) FTIR spectra of LNU-13 and LNU-13@I_2_.

## Data Availability

All data related to this study are presented in this publication.

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
