# Peer review of "A Carbazole-Functionalized Porous Aromatic Framework for Enhancing Volatile Iodine Capture via Lewis Electron Pairing"

_molecules, 2021, doi:10.3390/molecules26175263_

Round 1
Reviewer 1 Report
The authors report a novel porous aromatic framework based on 1,3,5-triethynylbenzene polymerized with functionalizing carbozole groups. The resulting framework was investigated in the context of iodine capture which is an important environmental issue e.g. due to the necessity to handle with the radioactive iodine production in the nuclear industry. The authors prove that the presented material exhibits a large iodine uptake that reaches thet highest level among the reported adsorbents. In addition, the perfect capture of iodine from the solution as well as the good reversibility of the iodine adsorption process. Taking the above into account, I recommend the publication of this work in the Molecules journal. Only minor revision is needed. This revision should take into account the following points:
- The critical point of the presented work is the achieved level of iodine uptake. As far as I understand from the experimental section, the measurement of the iodine adsorption was based on the simple measurements of the mass change after the exposition of the sample to the iodine vapors. The authors should clearly clarify the details of this method, its accuracy and the uncertainty of the resulting parameters of iodine uptake, in particular in comparison with the related values obtained for other materials investigated in this regard (Figure 4). It will be important to clarify if the identical method was used for the reported compound and for the other iodine adsorbents given in Fig. 4 to be sure that really the reported material is better than ALL reported iodine adsorbents.
- Even that the PXRD pattern shows the reported materials is basically amorphous, a few words based on the spectroscopic data should be given about the possible structural model (dimensionality, connectivity) of the material. The discussion on the possible structure based on the similar materials available in the literature will be also helpful.
Author Response
Point-to-point response to reviewer’s comments and corrections made in the revised manuscript
Referee 1:
Comment: The authors report a novel porous aromatic framework based on 1,3,5-triethynylbenzene polymerized with functionalizing carbozole groups. The resulting framework was investigated in the context of iodine capture which is an important environmental issue e.g. due to the necessity to handle with the radioactive iodine production in the nuclear industry. The authors prove that the presented material exhibits a large iodine uptake that reaches the highest level among the reported adsorbents. In addition, the perfect capture of iodine from the solution as well as the good reversibility of the iodine adsorption process. Taking the above into account, I recommend the publication of this work in the Molecules journal. Only minor revision is needed.
Response: Many thanks for his/her comments and appreciation of our work.
Question 1: The critical point of the presented work is the achieved level of iodine uptake. As far as I understand from the experimental section, the measurement of the iodine adsorption was based on the simple measurements of the mass change after the exposition of the sample to the iodine vapors. The authors should clearly clarify the details of this method, its accuracy and the uncertainty of the resulting parameters of iodine uptake, in particular in comparison with the related values obtained for other materials investigated in this regard (Figure 4). It will be important to clarify if the identical method was used for the reported compound and for the other iodine adsorbents given in Fig. 4 to be sure that really the reported material is better than ALL reported iodine adsorbents.
Response 1: Thanks for your suggestion. The comparison with the related values obtained for other materials was conducted by the same adsorption method. The experimental detail of the iodine adsorption measurement was listed in the revised manuscript:
Iodine adsorption from volatile iodine. Iodine absorption capacity was analyzed ac-cording to the gravimetric measurements. The LNU-13 powder (30.0 mg) was loaded into a small weighing bottle, which was then placed in a closed system at 348 K (75 ℃) and ambient pressure, along with excess non-radioactive solid iodine. After certain time intervals, the bottle was took out, cooled down to room temperature and weighted, and then loaded back into vapor of iodine to continue iodine absorption. The weight percentage of captured iodine was calculated using the following formula:
Adsorption capacity=(m2-m1)/m1 ×100 % (1)
Where m2 and m1 are the masses of PAF powder after and before iodine intake, respectively. (Please see Line 6, Page 7 and Line 24, Page 10 in the revised manuscript).
Question 2. Even that the PXRD pattern shows the reported materials is basically amorphous, a few words based on the spectroscopic data should be given about the possible structural model (dimensionality, connectivity) of the material. The discussion on the possible structure based on the similar materials available in the literature will be also helpful.
Response 2: Thanks for your suggestion. We made the change in the revised manuscript: Powder X-ray diffraction (XRD) pattern of LNU-13 shows a characteristic broad peak, indicating they are amorphous in nature. It seems that the formation of the stacked layer structure by the ordered connection among the building blocks is otherwise difficult [13,27]. (Please see Line 1, Page 3 in the revised manuscript).

Reviewer 2 Report
A carbazole-functionalized porous aromatic framework for enhancing volatile iodine capture via Lewis electron pairing
Zhuojun Yan,1 Bo Cui,1 Ting Zhao,2 Yifu Luo,2 Hongcui Zhang,1 Jialin Xie,1 Na Li,1 Naishun Bu,2,* Ye Yuan, 3,* and Lixin Xia1,4,*
The research work submitted for publication in Molecule journal by Yan et al. deals with iodine capture application using carbazole-based porous polymeric materials. Iodine is one of the pollutants produced during nuclear energy production and has a very long life. Hence, the treatment of pollutant before releasing to the atmosphere/water bodies becomes imperative. The strategy of utilizing the non-bonded electron lone pair present in the nitrogen attached with carbazole moiety imparts suitable bonding sites for iodine adsorption. Moreover, this also demonstrates that using hetero-atom functionalized building block helps achieve the uniform distribution of functional groups in porous framework, which many groups have exclusively explored for many other similar applications, e.g., J. Mater. Chem. A, 2017, 5, 258-265, Langmuir 34 (9), 2926-2932, J. Mater. Chem. A, 2017, 5, 21196 and Journal of CO2 Utilization 25, 302-309. The authors may use these references for improving the introduction to this research. Nevertheless, the authors have performed a thorough study, and the discussion is well supported by the data. The manuscript can be recommended for publication after addressing the below comments:
- The adsorption performance of the LNU-13 has primarily been compared with the powder-based adsorbent. The comparison of it with other forms of adsorbents like carbon foam (Chem. Eng. J., 382(2020), pp. 122833-122841), fibrous adsorbent (Ecotoxicology and Environmental Safety203 (2020): 111021), carbon cloth (Thin Solid Films 706 (2020): 138049) will make it more convenient for the readers to assess its sequestration performance.
- There are just three figures in the Supporting file, and one among these is TEM, which should be in the main manuscript. So, I don't feel the need for a separate supplementary information file and suggest the supplementary file figures can be moved to the main manuscript file.
Author Response
Point-to-point response to reviewer’s comments and corrections made in the revised manuscript
Referee: 2
Comment: The research work submitted for publication in Molecule journal by Yan et al. deals with iodine capture application using carbazole-based porous polymeric materials. Iodine is one of the pollutants produced during nuclear energy production and has a very long life. Hence, the treatment of pollutant before releasing to the atmosphere/water bodies becomes imperative. The strategy of utilizing the non-bonded electron lone pair present in the nitrogen attached with carbazole moiety imparts suitable bonding sites for iodine adsorption. Moreover, this also demonstrates that using hetero-atom functionalized building block helps achieve the uniform distribution of functional groups in porous framework, which many groups have exclusively explored for many other similar applications, e.g., J. Mater. Chem. A, 2017, 5, 258-265, Langmuir 34 (9), 2926-2932, J. Mater. Chem. A, 2017, 5, 21196 and Journal of CO2 Utilization 25, 302-309. The authors may use these references for improving the introduction to this research. Nevertheless, the authors have performed a thorough study, and the discussion is well supported by the data.
Response: Many thanks for his/her comments and appreciation of our work. We have cited these references for improving the introduction to this research. Please see References 11, 12, 17, and 19.
Question 1. The adsorption performance of the LNU-13 has primarily been compared with the powder-based adsorbent. The comparison of it with other forms of adsorbents like carbon foam (Chem. Eng. J., 382(2020), pp. 122833-122841), fibrous adsorbent (Ecotoxicology and Environmental Safety 203 (2020): 111021), carbon cloth (Thin Solid Films 706 (2020): 138049) will make it more convenient for the readers to assess its sequestration performance
Response 1: Thanks for your suggestion. We made the comparison with other forms of adsorbents in the revised manuscript:
Calculated by BET surface area (255 m2 g-1), the iodine uptake per specific surface area was ~ 8 mg m-2 which achieved the highest level among silver-containing zeolite [30], metal-organic frameworks (MOFs), and conjugated microporous polymers (CMPs), etc, reported by the same adsorption method, surpassing that of PAF-1 by ca. 30 times (Figure 4). Moreover, it also has a certain competitiveness compared with other forms of adsorbent, such as carbon foam, fiber adsorbent, carbon cloth, aerogel, etc. including BN foam (2.12 g g-1) [31], PE/PP-g-PNVP fibers (1.2378 g g-1) [32], C60-CC-PNP (2.4 g g-1) [33], CC-PNP (1.02 g g-1) [33], ENTDAT dried gel (1.8 g g-1) [34], G-TP5 (0.67 g g-1) [35] and G-TP6 (0.58 g g-1) [35]. (Please see Line 10, Page 7 in the revised manuscript)
Question 2. There are just three figures in the Supporting file, and one among these is TEM, which should be in the main manuscript. So, I don't feel the need for a separate supplementary information file and suggest the supplementary file figures can be moved to the main manuscript file.
Response 1: Thanks for your suggestion. We have moved the supplementary file figures to the main manuscript file. (Please see Fig. 2, Page 3 in the revised manuscript)
